# DIVERSITY-AWARE AGNOSTIC ENSEMBLE OF SHARPNESS MINIMIZERS

## ABSTRACT

There has long been a variety of theoretical and empirical evidence supporting the success of ensemble learning. Deep ensembles in particular leverage training randomness and expressivity of individual neural networks to gain prediction diversity and ultimately a boost in generalization performance, robustness and uncertainty estimation. In respect of generalization ability, it is found that minimizers pursuing wider local minima result in models being more robust to shifts between training and testing sets. A natural research question arises out of these two approaches as to whether better generalization ability can be achieved if ensemble learning and loss sharpness minimization is integrated. Our work takes the lead in investigating this connection and proposes DASH - a learning algorithm that promotes diversity and flatness within deep ensembles. More concretely, DASH encourages base learners to move divergently towards low-loss regions of minimal sharpness. We provide a theoretical backbone for our method along with empirical evidence demonstrating an improvement in ensemble generalization ability.

## 1 INTRODUCTION

Ensemble learning refers to learning a combination of multiple models in a way that the joint performance is better than than any of the ensemble members (so-called base learners). An ensemble can be an explicit collection of functionally independent models where the final decision is formed via approaches like averaging or majority voting of individual predictions. It can implicitly be a single model subject to stochastic perturbation of model architecture during training (Srivastava et al., 2014; Wan et al., 2013; Huang et al., 2016) or composed of sub-modules sharing some of the model parameters (Wenzel et al., 2020; Wen et al., 2020). An ensemble is called *homogeneous* if its base learners belong to the same model family and *heterogeneous* otherwise.

Traditional bagging technique (Breiman, 1996a) is shown to reduce variance among the base learners while boosting methods (Breiman, 1996b; Zhang & Zhang, 2008) are more likely to help reduce bias and improve generalization. Empirical evidence further points out that ensembles perform at least equally well as their base learners (Krogh & Vedelsby, 1994) and are much less fallible when the members are independently erroneous in different regions of the feature space (Hansen & Salamon, 1990). Deep learning models in particular often land at different local minima valleys due to training randomness, from initializations, mini-batch sampling, etc. This causes disagreement on predictions among model initializations given the same input. Meanwhile, deep ensembles (i.e., ensembles of deep neural networks) are found to be able to "smooth out" the highly non-convex loss surface, resulting in a better predictive performance (Hansen & Salamon, 1990; Perrone & Cooper, 1995; Garipov et al., 2018; Fort & Ganguli, 2019; Li et al., 2018). Ensemble models also benefit from the enhanced diversity in predictions, which is highlighted as another key driving force behind the success of ensemble learning (Dietterich, 2000). Further studies suggest that higher diversity among base learners leads to better robustness and predictive performance (Hansen & Salamon, 1990; Ovadia et al., 2019; Fort et al., 2019; Sinha et al., 2020). A recent work additionally shows that deep ensembles in general yield the best calibration under dataset shifts (Ovadia et al., 2019).

Tackling model generalization from a different approach, sharpness-aware minimization is an emerging line of work that seeks the minima within the flat loss regions, where SAM (Foret et al., 2021) is the most popular method. Flat minimizers have been theoretically and empirically proven in various applications to yield better testing accuracies (Jiang et al., 2020; Petzka et al., 2021; Dziugaite & Roy, 2017). At every training step, SAM performs one gradient ascent step to find the worst-case perturbations on the parameters. Given plenty of advantages of ensemble models, a natural question thus arises as to whether ensemble learning and sharpness-aware minimization can be

integrated to boost model generalization ability. In other words, *can we learn a deep ensemble of sharpness minimizers such that the entire ensemble is more generalizable?*

Motivated by this connection, our work proposes a sharpness-aware ensemble learning method that aims to improve the ensemble predictive accuracy. More concretely, we first develop a theory showing that the general loss of the ensemble can be reduced by minimizing loss sharpness for both the ensemble and its base learners (See Theorem 1). By encouraging the base learners to move closer to flat local minima, we however observe that under this sharpness-minimization scheme, the networks converge to low-loss tunnels that are close to one another, thereby compromising ensemble diversity (See Section 3.3). Fortunately, this means that ensemble generalization can still be further improved.

To this end, we contribute a novel diversity-aware agnostic term that navigates the individual learners to explore multiple wide minima in a divergent fashion. This term is introduced early on in the process where it first encourages individual learns to *agnostically* explore multiple potential gradient pathways, then to diverge towards those that help achieve the common goal.

In the remainder of the paper, we provide empirical evidence confirming that promoting diversity results in an increase the ensemble predictive performance and better uncertainty estimation capability than the baseline methods. To the best of our knowledge, we are the first to explore the connection between ensemble diversity and loss sharpness. Our work sheds lights on how to guide individual learners in a deep ensemble to collaborate effectively on a high-dimensional loss landscape. Our contributions in this paper are summarized as follows:

- We propose **DASH**: an ensemble learning method for **D**iversity-aware **A**gnostic Ensemble of **Sh**arpness Minimizers. **DASH** seeks to minimize generalization loss by directing the base classifiers in the ensemble towards diverse loss regions of maximal flatness.

- We provide a theoretical development for our method, followed by the technical insights into how adding the diversity-aware term promotes diversity in the ensemble that ultimately leads to better model generalization.

- Across various image classification tasks, we demonstrate an improvement in model generalization capacity of both homogeneous and heterogeneous ensembles up to 6%, in which the latter benefits significantly.

## 2 RELATED WORKS

**Ensemble Learning.** The rise of ensemble learning dates back to the development of classical techniques like bagging (Breiman, 1996a) or boosting (Breiman, 1996b; Freund et al., 1996; Friedman, 2001; Zhang & Zhang, 2008) for improving model generalization. While bagging algorithm involves training independent weak learners in parallel, boosting methods iteratively combine base learners to create a strong model where successor learners try to correct the errors of predecessor ones. In the era of deep learning, there has been an increase in attention towards ensembles of deep neural networks. A deep ensemble made up of low-loss neural learners has been consistently shown to yield to outperform an individual network (Hansen & Salamon, 1990; Perrone & Cooper, 1995; Huang et al., 2017; Garipov et al., 2018; Evci et al., 2020). In addition to predictive accuracy, deep ensembles has achieved successes in such other areas as uncertainty estimation (Lakshminarayanan et al., 2017; Ovadia et al., 2019; Gustafsson et al., 2020) or adversarial robustness (Pang et al., 2019; Kariyappa & Qureshi, 2019; Yang et al., 2021; 2020).

Ensembles often come with high training and testing costs that can grow linearly with the size of ensembles. This motivates recent works on efficient ensembles for reducing computational overhead without compromising their performance. One direction is to leverage the success of Dynamic Sparse Training (Liu et al., 2021; Mocanu et al., 2021; Evci et al., 2022) to generate an ensemble of sparse networks with lower training costs while maintaining comparable performance with dense ensembles (Liu et al., 2022). Another light-weight ensemble learning method is via pseudo or implicit ensembles that involves training a single model that exhibits the behavior or characteristic of an ensemble. Regularization techniques such as Drop-out (Srivastava et al., 2014; Gal & Ghahramani, 2016), Drop-connect (Wan et al., 2013) or Stochastic Depth (Huang et al., 2016) can be viewed as an ensemble network by masking the some units, connections or layers of the network. Other implicit strategies include training base learners with different hyperparameter configurations (Wenzel et al., 2020), decomposing the weight matrices into individual weight modules for each base learners (Wen et al., 2020) or using multi-input/output configuration to learn independent sub-networks within a single model (Havasi et al., 2020).

**Sharpness-Aware Minimization.** There has been a growing body of works that theoretically and empirically study the connection between loss sharpness and generalization capacity (Hochreiter & Schmidhuber, 1994; Neyshabur et al., 2017; Dinh et al., 2017; Fort & Ganguli, 2019). Convergence in flat regions of wider local minima has been found to improve out-of-distribution robustness of neural networks (Jiang et al., 2020; Petzka et al., 2021; Dziugaite & Roy, 2017). Some other works (Keskar et al., 2017; Jastrzebski et al., 2017; Wei et al., 2020) study the effect of the covariance of gradient or training configurations such as batch size, learning rate, dropout rate on the flatness of minima. One way to encourage search in flat minima is by adding regularization terms to the loss function such as Softmax output's low entropy penalty (Pereyra et al., 2017; Chaudhari et al., 2017) or distillation losses (Zhang et al., 2018; 2019).

SAM (Foret et al., 2021) is a recent flat minimizer widely known for its effectiveness and scalability, which encourages the model to search for parameters in the local regions that are uniformly low-loss. SAM has been actively exploited in various applications: meta-learning bi-level optimization in (Abbas et al., 2022), federated learning (Qu et al., 2022), domain generalization (Cha et al., 2021), multi-task learning (Phan et al., 2022) or for vision transformers (Chen et al., 2021) and language models (Bahri et al., 2022). Coming from two different directions, ensemble learning and sharpness-aware minimization yet share the same goal of improving generalization. Leveraging these two powerful learning strategies in a single framework remains an explored area. Our work contributes a novel effort to fill in this research gap.

## 3 METHODOLOGY

In this section, we first present the theoretical development demonstrating why incorporating sharpness awareness in ensemble learning is beneficial for improving the generalization ability of ensemble models. While encouraging sharpness in the base learners guides them closer towards flat regions of local minima, it compromises the ensemble diversity, which is crucial for ensemble learning. Addressing this issue, we later propose a novel early diversity-aware term that introduces diversity among the base learners.

### 3.1 ENSEMBLE SETTING AND NOTIONS

We now explain the ensemble setting and the notions used throughout our paper. Given $m$ base learners $f_{\theta_i}^{(i)}(x)$, $i = 1, ..., m$, we define the ensemble model

$$f_\theta^{\text{ens}}(x) = \frac{1}{m} \sum_{i=1}^m f_{\theta_i}^{(i)}(x),$$

where $\theta = [\theta_i]_{i=1}^m$, $x \in \mathbb{R}^d$, and $f(x) \in \Delta_{M-1} = \{\pi \in \mathbb{R}^M : \pi \geq 0 \wedge \|\pi\|_1 = 1\}$. Here, parameters $\theta_i$ and $\theta$ are used only for the classifier $f_{\theta_i}^{(i)}$ and the ensemble classifier $f_\theta^{\text{ens}}$, respectively. It is worth noting that the base learners $f_{\theta_i}^{(i)}$ can be different architectures.

Assume that $\ell : \mathbb{R}^M \times \mathcal{Y} \to \mathbb{R}$, where $\mathcal{Y} = [M] = \{1, \ldots, M\}$ is the label set, is a convex and bounded loss function. The training set is denoted by $\mathcal{S} = \{(x_i, y_i)\}_{i=1}^N$ of data points $(x_i, y_i) \sim \mathcal{D}$, where $\mathcal{D}$ is a data-label distribution. We denote

$$\mathcal{L}_\mathcal{S}(\theta_i) = \frac{1}{N} \sum_{j=1}^N \ell\left(f_{\theta_i}^i(x_j), y_j\right) \text{ and } \mathcal{L}_\mathcal{D}(\theta_i) = \mathbb{E}_{(x,y) \sim \mathcal{D}}\left[\ell(f_{\theta_i}^i(x), y)\right]$$

as the empirical and general losses w.r.t. the base learner $\theta_i$.

Similarly, we define the following empirical and general losses for the ensemble model

$$\mathcal{L}_\mathcal{S}(\theta) = \frac{1}{N} \sum_{j=1}^N \ell\left(f_\theta^{\text{ens}}(x_j), y_j\right) \text{ and } \mathcal{L}_\mathcal{D}(\theta) = \mathbb{E}_{(x,y) \sim \mathcal{D}}\left[\ell\left(f_\theta^{\text{ens}}(x), y\right)\right].$$

One of the key motivations is to ensure the ensemble model $f_\theta^{ens}$ can generalize well on the data-label distribution $\mathcal{D}$, while only be optimized on a finite training set $\mathcal{S}$. To achieve this desideratum, we develop an upper-bound for the general loss of the ensemble model in Section 3.2.

## 3.2 SHARPNESS-AWARE ENSEMBLE LEARNING

The following theorem explains how minimizing the sharpness for the ensemble and base learners help reduce the general loss of the ensemble model, whose proof can be found in Appendix A.

**Theorem 1.** *Assume that the loss function $\ell$ is convex and upper-bounded by $L$. With the probability at least $1 - \delta$ over the choices of $\mathcal{S} \sim \mathcal{D}^N$, for any $0 \leq \gamma \leq 1$, we have*

$$
\mathcal{L}_{\mathcal{D}}\left(\theta\right) \leq \frac{(1-\gamma)}{m} \sum_{i=1}^{m} \max_{\theta_i': \|\theta_i' - \theta_i\| < \rho} \mathcal{L}_{\mathcal{S}}\left(\theta_i'\right) + \gamma \max_{\theta': \|\theta' - \theta\| < \sqrt{m}\rho} \mathcal{L}_{\mathcal{S}}(\theta') + \frac{CL}{\sqrt{N}} \times \left\{ m\sqrt{\log \frac{mNk}{\delta}} \right.
$$

$$
+ \sqrt{km \log \left(1 + \frac{\sum_{i=1}^{m} \|\theta_i\|^2}{m\rho^2} \left(1 + \sqrt{\log(N)/(mk)}\right)^2\right)}
$$

$$
\left. + \sum_{i=1}^{m} \sqrt{k \log \left(1 + \frac{\|\theta_i\|^2}{\rho^2} \left(1 + \sqrt{\log(N)/k}\right)^2\right)} + O(1) \right\},
$$

*where $C$ is a universal constant.*

As introduced in Foret et al. (2021), given a model $f_\theta$, the sharpness is defined as the maximum loss difference between the model and its perturbed version, i.e., $\max_{\theta': \|\theta' - \theta\| < \rho} \mathcal{L}_{\mathcal{S}}\left(\theta'\right) - \mathcal{L}_{\mathcal{S}}\left(\theta\right)$ which is an upper bound of the generalization error $\mathcal{L}_{\mathcal{D}}(\theta)$. Therefore, minimizing the sharpness can help to improve the generalization ability of the model. However, most of the previous sharpness-aware methods focused on a single model. For the first time, we connect the sharpness-aware minimization with ensemble learning. Theorem 1 suggests that enforcing sharpness awareness for both the ensemble and base learners could assist us in improving the generalization ability of the ensemble model. More specifically, the first term on the RHS of the inequality can be interpreted as the average sharpness of the base learners, while the second term is the sharpness of the ensemble model. The trade-off parameter $\gamma$ signifies the levels of sharpness-aware enforcement for the ensemble model alone and its base learners themselves.

To further investigate the trade-off between these terms, we conduct the experiments on the CIFAR100 dataset by varying $\gamma$ and observing the ensemble performance as shown in Figure 1. It can be seen that varying $\gamma$ does significantly affect the ensemble performance, with a difference of more than 1.8% in ensemble accuracy. Interestingly, the ensemble accuracy and its uncertainty estimation capability peak when $\gamma = 0$ and decrease when $\gamma$ increases. This observation suggests that minimizing the sharpness of the base learners is more effective than minimizing the sharpness of the ensemble model, which is the base setting of our method from now on. This antagonistic behavior of the ensemble model interestingly concurs with an observation in Allen-Zhu & Li (2022). The paper shows that directly training an average of neural learners in fact barely yields any better performance than the individual networks because the learners end up learning the same set of features. In our case, increasing $\gamma$ may adversely induce this behavior. However, this motivates the intuition that diversity in the ensemble will play an important role in the learning process, especially in the multi-view data setting introduced in Allen-Zhu & Li (2022) commonly found in classification tasks.

## 3.3 DIVERSITY-AWARE AGNOSTIC ENSEMBLE OF FLAT BASE LEARNERS

We now dig into the technical details of sharpness-aware ensemble learning discussed in the previous section to clearly understand the ensemble behavior. If we enforce the sharpness within the base learners with SAM as in Foret et al. (2021), a given model $f_{\theta_i}^i$ is updated as

$$
\theta_i^a = \theta_i + \rho_1 \frac{\nabla_{\theta_i} \mathcal{L}_B\left(\theta_i\right)}{\|\nabla_{\theta_i} \mathcal{L}_B\left(\theta_i\right)\|}, \tag{1}
$$

$$
\theta_i = \theta_i - \eta \nabla_{\theta_i} \mathcal{L}_B\left(\theta_i^a\right).
$$

where $B$ is the current mini-batch, $\rho_1 > 0$ is the perturbed radius, and $\eta > 0$ is the learning rate.

To inspect the behavior of the above updates, let us analyze the following gradient using the first-order Taylor expansion

$$
\nabla_{\theta_i} \mathcal{L}_B\left(\theta_i^a\right) = \nabla_{\theta_i} \left[\mathcal{L}_B\left(\theta_i + \rho_1 \frac{\nabla_{\theta_i} \mathcal{L}_B\left(\theta_i\right)}{\|\nabla_{\theta_i} \mathcal{L}_B\left(\theta_i\right)\|}\right)\right]
$$

$$
\approx \nabla_{\theta_i} \left[\mathcal{L}_B\left(\theta_i\right) + \rho_1 \nabla_{\theta_i} \mathcal{L}_B\left(\theta_i\right) \cdot \frac{\nabla_{\theta_i} \mathcal{L}_B\left(\theta_i\right)}{\|\nabla_{\theta_i} \mathcal{L}_B\left(\theta_i\right)\|}\right] = \nabla_{\theta_i} \left[\mathcal{L}_B\left(\theta_i\right) + \rho_1 \|\nabla_{\theta_i} \mathcal{L}_B\left(\theta_i\right)\|\right], \tag{2}
$$

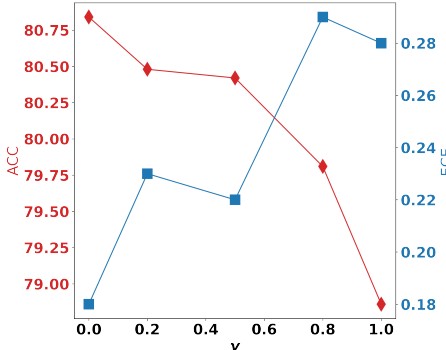

Figure 1: Tuning parameter $\gamma$. Both the ensemble accuracy (higher is better) and the expected calibration error (ECE, lower is better) peak when $\gamma = 0$. See Table 12 for other metrics.

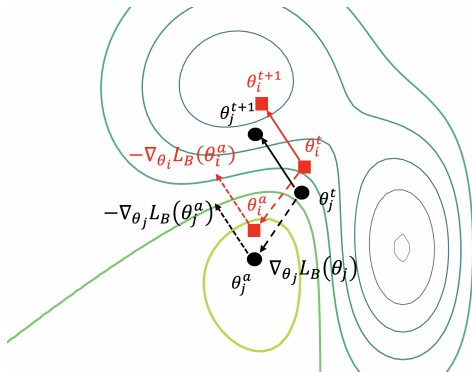

Figure 2: Illustration of the model dynamics under sharpness-aware term on loss landscape. Two base learners $\theta_i$ and $\theta_j$ (represented by the red and black vectors respectively) happen to be initialized closely. At each step, since updated independently yet using the same mini-batch from $\theta_i$ and $\theta_j$, two perturbed models $\theta_i^a$ and $\theta_i^a$ are less diverse, hence two updated models $\theta_i$ and $\theta_j$ are also less diverse and more likely end up at the same low-loss and flat region.

where $\cdot$ represents the dot product.

The approximation in (2) indicates that since we follow the negative gradient $-\nabla_{\theta_i} \mathcal{L}_B (\theta_i^a)$ when updating the current model $\theta_i$, the new model tends to decrease both the loss $\mathcal{L}_B (\theta_i)$ and the gradient norm $\|\nabla_{\theta_i} \mathcal{L}_B (\theta_i)\|$, directing the base learners to go into the low-loss and flat regions as expected. However, it can be seen that the current mechanism lacks cooperation among the base learners, which possibly reduces the diversity among them. This may stem from the usage of well-known initialization techniques (e.g., He initializer (He et al., 2015) or Xavier initializer (Glorot & Bengio, 2010)), making the initial base models $\theta_i, i = 1, \ldots, m$ significantly less diverse. Moreover, the normalized gradients $\frac{\nabla_{\theta_i} \mathcal{L}_B(\theta_i)}{\|\nabla_{\theta_i} \mathcal{L}_B(\theta_i)\|}, i = 1, \ldots, m$ reveals that the perturbed models $\theta_i^a, i = 1, \ldots, m$ are also less diverse because they are computed using the same mini-batch $B$. This eventually leads to less diverse updated models $\theta_i, i = 1, \ldots, m$, which is illustrated in Figure 2.

It is natural to ask the question: *"how to strengthen the update in Eq. (1) that encourages the base learners to be more diverse, while still approaching their low-loss and flat regions."* To this end, we propose the following "agnostic" update approach method so that the desired properties are explicitly achieved after updating

$$\theta_i^a = \theta_i + \rho_1 \frac{\nabla_{\theta_i} \mathcal{L}_B (\theta_i)}{\|\nabla_{\theta_i} \mathcal{L}_B (\theta_i)\|} + \rho_2 \frac{\nabla_{\theta_i} \mathcal{L}_B^{div} (\theta_i, \theta_{\neq i})}{\|\nabla_{\theta_i} \mathcal{L}_B^{div} (\theta_i, \theta_{\neq i})\|},$$

$$\theta_i = \theta_i - \eta \nabla_{\theta_i} \mathcal{L}_B (\theta_i^a),$$

$$(3)$$

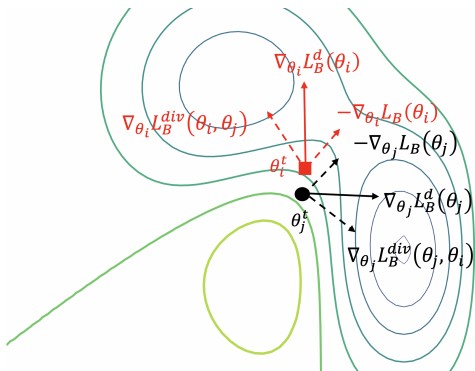

Figure 3: Illustration of the model dynamics under diversity-aware term. Given two base learners $\theta_i$ and $\theta_j$ (represented by the red and black vectors respectively), the gradients $-\nabla_{\theta_i}\mathcal{L}_B(\theta_i)$ and $-\nabla_{\theta_i}\mathcal{L}_B(\theta_i)$ navigate the models towards their low-loss (also flat) regions. Moreover, the two gradients $\nabla_{\theta_i}\mathcal{L}_B^{div}(\theta_i,\theta_{\neq i})$ and $\nabla_{\theta_j}\mathcal{L}_B^{div}(\theta_j,\theta_{\neq j})$ encourage the models to move divergently. As discussed, our update strategy forces the two gradients $-\nabla_{\theta_i}\mathcal{L}_B(\theta_i)$ and $\nabla_{\theta_i}\mathcal{L}_B^{div}(\theta_i,\theta_{\neq i})$ to be more congruent. As the result, two models are divergently oriented to two non-overlapping low-loss and flat regions. This behavior is imposed similarly for the other pair w.r.t. the model $\theta_j$, altogether enhancing the ensemble diversity.

where $\theta_{\neq i}$ specifies the set of models excluding $\theta_i$ and the $i$-th divergence loss is defined as

$$\mathcal{L}_B^{div}(\theta_i,\theta_{\neq i}) = \frac{1}{|B|}\sum_{(x,y)\in B}\sum_{j\neq i}\text{KL}\left(\sigma\left(\frac{h_{\theta_i}^i(x)}{\tau}\right),\sigma\left(\frac{h_{\theta_j}^j(x)}{\tau}\right)\right), \qquad (4)$$

where $h_{\theta_k}^k$ returns *non-targeted logits* (i.e., excluding the logit value of the ground-truth class) of the $k$-th base learner, $\sigma$ is the softmax function, $\tau > 0$ is the temperature variable, $\rho_2$ is another perturbed radius, and KL specifies the Kullback-Leibler divergence. In practice, we choose $\rho_2 = \rho_1$ for simplicity and $\tau < 1$ to favor the distance on dominating modes on each base learner.

It is worth noting that in the formulation of the divergence loss in Eq. (4), we only use the *non-targeted logits* to diversify the non-targeted parts. The reason is that we aim to diversify the base learners without interfering the their performance on predicting ground-truth labels.

To inspect the agnostic behavior of the second gradient when adding to the formula of the perturbed models $\theta_i^a$, we investigate the following gradient using the first-order Taylor expansion

$$\begin{aligned}
\nabla_{\theta_i}\mathcal{L}_B(\theta_i^a) =& \nabla_{\theta_i}\left[\mathcal{L}_B\left(\theta_i + \rho_1\frac{\nabla_{\theta_i}\mathcal{L}_B(\theta_i)}{\|\nabla_{\theta_i}\mathcal{L}_B(\theta_i)\|} + \rho_2\frac{\nabla_{\theta_i}\mathcal{L}_B^{div}(\theta_i,\theta_{\neq i})}{\|\nabla_{\theta_i}\mathcal{L}_B^{div}(\theta_i,\theta_{\neq i})\|}\right)\right] \\
\approx& \nabla_{\theta_i}\left[\mathcal{L}_B(\theta_i) + \rho_1\nabla_{\theta_i}\mathcal{L}_B(\theta_i)\cdot\frac{\nabla_{\theta_i}\mathcal{L}_B(\theta_i)}{\|\nabla_{\theta_i}\mathcal{L}_B(\theta_i)\|} + \rho_2\nabla_{\theta_i}\mathcal{L}_B(\theta_i)\cdot\frac{\nabla_{\theta_i}\mathcal{L}_B^{div}(\theta_i,\theta_{\neq i})}{\|\nabla_{\theta_i}\mathcal{L}_B^{div}(\theta_i,\theta_{\neq i})\|}\right] \\
=& \nabla_{\theta_i}\left[\mathcal{L}_B(\theta_i) + \rho_1\|\nabla_{\theta_i}\mathcal{L}_B(\theta_i)\| - \rho_2\frac{-\nabla_{\theta_i}\mathcal{L}_B(\theta_i)\cdot\nabla_{\theta_i}\mathcal{L}_B^{div}(\theta_i,\theta_{\neq i})}{\|\nabla_{\theta_i}\mathcal{L}_B^{div}(\theta_i,\theta_{\neq i})\|}\right]. \qquad (5)
\end{aligned}$$

In Eq. (5), the first two terms lead the base learners to go to their low-loss and flat regions as discussed before. We then analyze the agnostic behavior of the third term. According to the update formula of $\theta_i$ in Eq. (3), we follow the positive direction of $\nabla_{\theta_i}\mathcal{L}_{\mathcal{B}}^d = \nabla_{\theta_i}\left[\frac{-\nabla_{\theta_i}\mathcal{L}_B(\theta_i)\cdot\nabla_{\theta_i}\mathcal{L}_B^{div}(\theta_i,\theta_{\neq i})}{\|\nabla_{\theta_i}\mathcal{L}_B^{div}(\theta_i,\theta_{\neq i})\|}\right]$, further implying that the updated base learner networks aim to maximize $\frac{-\nabla_{\theta_i}\mathcal{L}_B(\theta_i)\cdot\nabla_{\theta_i}\mathcal{L}_B^{div}(\theta_i,\theta_{\neq i})}{\|\nabla_{\theta_i}\mathcal{L}_B^{div}(\theta_i,\theta_{\neq i})\|}$. Therefore, the low-loss direction $-\nabla_{\theta_i}\mathcal{L}_B(\theta_i)$ becomes more congruent with $\frac{\nabla_{\theta_i}\mathcal{L}_B^{div}(\theta_i,\theta_{\neq i})}{\|\nabla_{\theta_i}\mathcal{L}_B^{div}(\theta_i,\theta_{\neq i})\|}$, meaning that the base learners tend to diverge while moving along the low-loss and flat directions. Figure 3 visualizes our arguments.

Table 1: Evaluation of the ensemble **accuracy** (%) on the CIFAR10/100 and Tiny-ImageNet datasets. R10x5 indicates an ensemble of five ResNet10 models. R18x3 indicates an ensemble of three ResNet18 models. RME indicates an ensemble of ResNet18, MobileNet and EfficientNet, respectively.

| Accuracy ↑ | CIFAR10 | | | CIFAR100 | | | Tiny-ImageNet |
| | R10x5 | R18x3 | RME | R10x5 | R18x3 | RME | R18x3 |
|---|---|---|---|---|---|---|---|
| Deep Ensemble | 92.7 | 93.7 | 89.0 | 73.7 | 75.4 | 62.7 | 65.9 |
| Fast Geometric | 92.5 | 93.3 | - | 63.2 | 72.3 | - | 61.8 |
| Snapshot | 93.6 | 94.8 | - | 72.8 | 75.7 | - | 62.2 |
| EDST | 92.0 | 92.8 | - | 68.4 | 69.6 | - | 62.3 |
| DST | 93.2 | 94.7 | 93.4 | 70.8 | 70.4 | 71.7 | 61.9 |
| SGD | 95.1 | 95.2 | 92.6 | 75.9 | 78.9 | 72.6 | 62.3 |
| SAM | 95.4 | 95.8 | 93.8 | 77.7 | 80.1 | 76.4 | 66.1 |
| DASH (Ours) | **95.7** | **96.7** | **95.2** | **80.8** | **82.2** | **78.7** | **69.9** |

Table 2: Evaluation of Uncertainty Estimation (UE). **Calibrated-Brier score** is chosen as the representative UE metric reported in this table. Evaluation on all **six** UE metrics for CIFAR10/100 can be found in the supplementary material. Overall, our method achieves better calibration than baselines on several metrics, especially in the heterogeneous ensemble setting.

| Cal-Brier ↓ | CIFAR10 | | | CIFAR100 | | | Tiny-ImageNet |
| | R10x5 | R18x3 | RME | R10x5 | R18x3 | RME | R18x3 |
|---|---|---|---|---|---|---|---|
| Deep Ensemble | 0.091 | 0.079 | 0.153 | 0.329 | 0.308 | 0.433 | 0.453 |
| Fast Geometric | 0.251 | 0.087 | - | 0.606 | 0.344 | - | 0.499 |
| Snapshot | 0.083 | 0.071 | - | 0.338 | 0.311 | - | 0.501 |
| EDST | 0.122 | 0.113 | - | 0.427 | 0.412 | - | 0.495 |
| DST | 0.102 | 0.083 | 0.102 | 0.396 | 0.405 | 0.393 | 0.500 |
| SGD | 0.078 | 0.076 | 0.113 | 0.346 | 0.304 | 0.403 | 0.518 |
| SAM | 0.073 | 0.067 | 0.094 | 0.321 | 0.285 | 0.347 | 0.469 |
| DASH (Ours) | **0.067** | **0.056** | **0.075** | **0.267** | **0.255** | **0.298** | **0.407** |

## 4 EXPERIMENTS

We evaluate our methods on the classification tasks on CIFAR10/100 and Tiny-Imagenet. We experiment with homogeneous ensembles wherein all base learners has the same model architecture, i.e., R18x3 is an ensemble which consists of three ResNet18 models. We also experiment with heterogeneous ensemble, i.e., RME is an ensemble which consists of ResNet18, MobileNet and EfficientNet models. The configuration shared between our method and the baselines involves model training for 200 epochs using SGD optimizer with weight decay of 0.005. We follow the standard data preprocessing schemes that consists of zero-padding with 4 pixels on each side, random crop, horizon flip and normalization. The ensemble prediction has been aggregated by averaging the softmax predictions of all base classifiers.[1] In all tables, bold/underline indicates the best/second-best method. ↑,↓ respectively indicates higher/lower performance is better

### 4.1 BASELINES

This work focuses on improving generalization of ensembles. We compare our method against top ensemble methods with high predictive accuracies across literature: Deep ensembles (Lakshminarayanan et al., 2017), Snapshot ensembles (Huang et al., 2017), Fast Geometric Ensemble (FGE) (Garipov et al., 2018), sparse ensembles EDST and DST (Liu et al., 2022). We also deploy SGD and SAM (Foret et al., 2021) as different optimizers to train an ensemble model and consider as two additional baselines to compare with.

### 4.2 METRICS

We use Ensemble accuracy (Acc) as the primary metric used to measure the generalization of an ensemble learning method. To evaluate the uncertainty capability of a model, we use the standard metrics: Negative Log-Likelihood (NLL), Brier score, and Expected Calibration Error (ECE), which are widely used in the literature. Additionally, we employ calibrated uncertainty estimation (UE)

---

[1]Our code is anonymously published at https://anonymous.4open.science/r/DASH.

metrics, such as Cal-NLL, Cal-Brier, and Cal-AAC, at the optimal temperature to avoid measuring calibration error that can be eliminated by simple temperature scaling, as suggested in Ashukha et al. (2020). To measure ensemble diversity, we use Disagreement (D) of predictions, which is a common metric (Kuncheva & Whitaker, 2003). We also utilize the Log of Determinant (LD) of a matrix consisting of non-target predictions of base classifiers, as proposed in Pang et al. (2019). The LD metric provides an elegant geometric interpretation of ensemble diversity, which is better than the simple disagreement metric.

## 4.3 EVALUATION OF PREDICTIVE PERFORMANCE

The results presented in Table 1 demonstrate the effectiveness of our proposed method, DASH, in improving the generalization ability of ensemble methods. Across all datasets and architectures, DASH consistently and significantly outperformed all baselines. For example, when compared to SGD with R18x3 architecture, DASH achieved substantial improvement gaps of 1.5%, 3.3%, and 7.6% on the CIFAR10, CIFAR100, and Tiny-ImageNet datasets, respectively. When compared to Deep Ensemble, DASH achieved improvement gaps of 3.0%, 6.8%, and 4.0%, respectively, on these same datasets. Our results also provide evidence that seeking more flat classifiers can bring significant benefits to ensemble learning. SAM achieves improvements over SGD or Deep Ensemble, but DASH achieved even greater improvements. Specifically, on the CIFAR100 dataset, DASH outperformed SAM by 3.1%, 2.1%, and 2.3% with R10x5, R18x3, and RME architectures, respectively, while that improvement on the Tiny-ImageNet dataset was 3.8%. This improvement indicates the benefits of effectively collaborating between flatness and diversity seeking objectives in deep ensembles. Unlike Fast Geometric, Snapshot, or EDST methods, which are limited to homogeneous ensemble settings, DASH is a general method capable of improving ensemble performance even when ensembling different architectures. This is evidenced by the larger improvement gaps over SAM on the RME architecture (i.e., 1.4% improvement on the CIFAR10 dataset) compared to the R18x3 architecture (i.e., 0.9% improvement on the same dataset). These results demonstrate the versatility and effectiveness of DASH in improving the generalization ability of deep ensembles across diverse architectures and datasets.

## 4.4 EVALUATION OF UNCERTAINTY ESTIMATION

Although improving uncertainty estimation is not the primary focus of our method, in this section we still would like to investigate the effectiveness of our method on this aspect by measuring six UE metrics across all experimental settings. We present the results of our evaluation in Table 2, where we compare the uncertainty estimation capacity of our method with various baselines using the Calibrated-Brier score as the representative metric. Our method consistently achieves the best performance over all baselines across all experimental settings. For instance, on the CIFAR10 dataset with the R10x5 setting, our method obtains a score of 0.067, a relative improvement of 26% over the Deep Ensemble method. Similarly, across all settings, our method achieves a relative improvement of 26%, 29%, 51%, 18%, 17%, 31%, and 10% over the Deep Ensemble method. Furthermore, in Table 3, we evaluate the performance of our method on all six UE metrics on the Tiny-ImageNet dataset. In this setting, our method achieves the best performance on five UE metrics, except for the ECE metric. Compared to the Deep Ensemble method, our method obtains a relative improvement of 1%, 1%, 10%, 3%, and 14% on the NLL, Brier, Cal-Brier, Cal-ACC, and Cal-NLL metrics, respectively. In conclusion, our method shows promising results in improving uncertainty estimation, as demonstrated by its superior performance in various UE metrics.

## 5 ABLATION STUDIES

### 5.1 EFFECT OF SHARPNESS-AWARE MINIMIZATION

Since proposed in Foret et al. (2021), there are several sharpness-aware minimization methods have been developed to address various limitations of the pioneer method. Notably, Kwon et al. (2021) proposed an adaptive method to reduce the sensitivity to parameter re-scaling issue, thus reducing the gap between sharpness and generalization of a model. In this section, we would like to examine the impact of different sharpness-aware methods to the final performance when integrating into our method. More specifically, we consider two sharpness-aware methods which are Standard (Non-Adaptive) SAM (Foret et al., 2021) and Adaptive SAM (Kwon et al., 2021), corresponding to our two variants which are Standard DASH and Adaptive DASH. We conduct experiment on the CIFAR10 and CIFAR100 datasets with two ensemble settings, i.e., R18x3 and R10x5 architectures and report results in Table 4. We choose $\rho = 0.05$ for the standard version and $\rho = 2.0$ for the adaptive version

Table 3: Evaluation of Uncertainty Estimation (UE) across six standard UE metrics on the Tiny-ImageNet dataset with R18x3 architecture.

| | NLL ↓ | Brier ↓ | ECE ↓ | Cal-Brier ↓ | Cal-AAC ↓ | Cal-NLL ↓ |
|---|---|---|---|---|---|---|
| Deep Ensemble | 1.400 | 0.452 | **0.110** | 0.453 | 0.210 | 1.413 |
| Fast Geometric | 1.548 | 0.501 | 0.116 | 0.499 | 0.239 | 1.544 |
| Snapshot | 1.643 | 0.505 | 0.118 | 0.501 | 0.237 | 1.599 |
| EDST | 1.581 | 0.496 | 0.115 | 0.495 | 0.235 | 1.548 |
| DST | 1.525 | 0.499 | **0.110** | 0.500 | 0.239 | 1.536 |
| SGD | 1.999 | 0.601 | 0.283 | 0.518 | 0.272 | 1.737 |
| SAM | 1.791 | 0.563 | 0.297 | 0.469 | 0.242 | 1.484 |
| DASH (Ours) | **1.379** | **0.447** | 0.184 | **0.407** | **0.204** | **1.213** |

Table 4: Analysis of the effect of the sharpness aware methods on the CIFAR10 (C10) and CIFAR100 (C100) datasets. A denotes the adaptive sharpness-aware minimization, which is scale-invariant as proposed in. S denotes the standard (non adaptive) version.

| | | R18x3 | | R10x5 | |
|---|---|---|---|---|---|
| | | Accuracy ↑ | Cal-Brier ↓ | Accuracy ↑ | Cal-Brier ↓ |
| | SGD | 95.2 | 0.076 | 95.0 | 0.078 |
| | SAM | 95.8 | 0.067 | 95.4 | 0.073 |
| C10 | S-DASH | 96.4 | 0.060 | **95.8** | **0.066** |
| | A-SAM | 96.0 | 0.064 | 95.6 | 0.071 |
| | A-DASH | **96.7** | **0.056** | 95.7 | 0.067 |
| | SGD | 78.9 | 0.304 | 75.8 | 0.346 |
| | SAM | 80.1 | 0.285 | 77.7 | 0.321 |
| C100 | S-DASH | 81.7 | 0.262 | 80.3 | 0.279 |
| | A-SAM | 80.9 | 0.275 | 79.1 | 0.301 |
| | A-DASH | **82.2** | **0.255** | **80.9** | **0.267** |

as suggested in the project [2]. The results show that integrating sharpness-aware methods into our approach yields a significant improvement, regardless of the version. For example, Adaptive-DASH outperforms Adaptive-SAM across all settings, in both generalization and uncertainty estimation capability. Notably, the improvements are 1.3% and 1.76% in CIFAR100 dataset prediction tasks with R18x3 and R10x5 architectures, respectively. Similarly, Standard-DASH achieves a significant improvement over Standard-SAM in all settings, with the highest improvement being 2.55% ensemble accuracy with the R10x5 architecture on the CIFAR100 dataset. Interestingly, our Standard-DASH version even outperforms the Adaptive-SAM to achieve the second-best performance, just after our Adaptive-DASH version. This result emphasizes the effectiveness and generality of our method in various settings. Based on these results, we use the Adaptive-DASH as the default setting.

## 6 CONCLUSION

We developed **DASH** Ensemble - a learning algorithm that optimizes for deep ensembles of diverse and flat minimizers. Our method begins with a theoretical development to minimize sharpness-aware upper bound for the general loss of the ensemble, followed by a novel addition of an agnostic term to promote divergence among base classifiers. Our experimental results support the effectiveness of the agnostic term in introducing diversity in individual predictions, which ultimately leads to an improvement in generalization performance. This work has demonstrated the potential of integrating sharpness-aware minimization technique into the ensemble learning paradigm. We thus hope to motivate future works to exploit such a connection to develop more powerful and efficient models.

---

[2]https://github.com/davda54/sam

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

## A  PROOFS

**Theorem 2.** *Assume that the loss function $\ell$ is convex and upper-bounded by $L$. With the probability at least $1 - \delta$ over the choices of $\mathcal{S} \sim \mathcal{D}^N$, for any $0 \leq \gamma \leq 1$, we have*

$$\mathcal{L}_{\mathcal{D}}(\theta) \leq \gamma \max_{\theta': \|\theta' - \theta\| \leq \sqrt{m}\rho} \mathcal{L}_{\mathcal{S}}(\theta') + \frac{1 - \gamma}{m} \Big[ \sum_{i=1}^{m} \max_{\theta'_i: \|\theta'_i - \theta_i\| \leq \rho} \mathcal{L}_{\mathcal{S}}(\theta'_i) \Big] + \frac{CL}{\sqrt{N}} \times$$

$$\Big[ m \sqrt{\log \frac{m(N+k)}{\delta}} + \sum_{i=1}^{m} \sqrt{k \log \Big( 1 + \frac{\|\theta_i\|^2}{\rho^2} (1 + \sqrt{\log(N)}/k)^2 \Big)} +$$

$$\sqrt{km \log \Big( 1 + \frac{\sum_{i=1}^{m} \|\theta_i\|^2}{m\rho^2} \big( 1 + \sqrt{\log(N)/(mk)} \big)^2 \Big)} + O(1) \Big]$$

*where the $\theta_i$ and the loss function $\ell$ satisfying the conditions: for all $\rho > 0$, $P_i \sim \mathcal{N}(\theta_i, \rho^2 \mathbb{I}_k)$ and $P = \mathcal{N}(\theta, \rho^2 \mathbb{I}_{mk})$*

$$\mathbb{E}_{(x,y)\in\mathcal{D}} \big[ \ell(f_{\theta_i}^i(x), y) \big] \leq \mathbb{E}_{\theta'_i \sim P_i} \mathbb{E}_{(x,y)\in\mathcal{D}} \big[ \ell(f_{\theta'_i}^i(x), y) \big]$$

$$\mathbb{E}_{(x,y)\in\mathcal{D}} \big[ \ell(f_{\theta}^{\text{ens}}(x), y) \big] \leq \mathbb{E}_{\theta \sim P} \mathbb{E}_{(x,y)\in\mathcal{D}} \big[ \ell(f_{\theta}^{\text{ens}}(x), y) \big]$$

*Proof.* We use the PAC-Bayes theory in this proof. In PAC-Bayes theory, $\theta$ could follow a distribution, says $P$, thus we define the expected loss over $\theta$ distributed by $P$ as follows:

$$\mathcal{L}_{\mathcal{D}}(\theta, P) = \mathbb{E}_{\theta \sim P}[\ell_{\mathcal{D}}(\theta)]$$
$$\mathcal{L}_{\mathcal{S}}(\theta, P) = \mathbb{E}_{\theta \sim P}[\ell_{\mathcal{S}}(\theta)].$$

For any distribution $P = \mathcal{N}(\mathbf{0}, \sigma_P^2 \mathbb{I}_k)$ and $Q = \mathcal{N}(\theta, \sigma^2 \mathbb{I}_k)$ over $\theta \in \mathbb{R}^k$, where $P$ is the prior distribution and $Q$ is the posterior distribution, use the PAC-Bayes theorem in Alquier et al. (2016), for all $\beta > 0$, with a probability at least $1 - \delta$, we have

$$\mathcal{L}_{\mathcal{D}}(\theta, Q) \leq \mathcal{L}_{\mathcal{S}}(\theta, Q) + \frac{1}{\beta}\Big[\mathsf{KL}(Q\|P) + \log\frac{1}{\delta} + \Psi(\beta, N)\Big], \tag{6}$$

where $\Psi$ is defined as

$$\Psi(\beta, N) = \log \mathbb{E}_P \mathbb{E}_{\mathcal{D}^N}\Big[\exp\big\{\beta[\mathcal{L}_{\mathcal{D}}(f_\theta) - \mathcal{L}_{\mathcal{S}}(f_\theta)]\big\}\Big].$$

When the loss function is bounded by $L$, then

$$\Psi(\beta, N) \leq \frac{\beta^2 L^2}{8N}.$$

The task is to minimize the second term of RHS of equation 6, we thus choose $\beta = \sqrt{8N}\frac{\mathsf{KL}(Q\|P) + \log\frac{1}{\delta}}{L}$. Then the second term of RHS of equation 6 is equal to

$$\sqrt{\frac{\mathsf{KL}(Q\|P) + \log\frac{1}{\delta}}{2N}} \times L.$$

The KL divergence between $Q$ and $P$, when they are Gaussian, is given by formula

$$\mathsf{KL}(Q\|P) = \frac{1}{2}\left[\frac{k\sigma^2 + \|\theta\|^2}{\sigma_P^2} - k + k\log\frac{\sigma_P^2}{\sigma^2}\right].$$

For given posterior distribution $Q$ with fixed $\sigma^2$, to minimize the KL term, the $\sigma_P^2$ should be equal to $\sigma^2 + \|\theta\|^2/k$. In this case, the KL term is no less than

$$k\log\left(1 + \frac{\|\theta_0\|^2}{k\sigma^2}\right).$$

Thus, the second term of RHS is

$$\sqrt{\frac{\mathsf{KL}(Q\|P) + \log\frac{1}{\delta}}{2N}} \times L \geq \sqrt{\frac{k\log\left(1 + \frac{\|\theta\|^2}{k\sigma^2}\right)}{4N}} \times L \geq L$$

when $\|\theta\|^2 > \sigma^2\{\exp(4N/k) - 1\}$. Hence, for any $\|\theta\|_2 > \sigma^2\{\exp(4N/k) - 1\}$, we have the RHS is greater than the LHS, the inequality is trivial. In this work, we only consider the case:

$$\|\theta\|^2 < \sigma^2\big(\exp\{4N/k\} - 1\big). \tag{7}$$

Distribution $P$ is Gaussian centered around $\mathbf{0}$ with variance $\sigma_P^2 = \sigma^2 + \|\theta\|^2/k$, which is unknown at the time we set up the inequality, since $\theta$ is unknown. Meanwhile we have to specify $P$ in advance, since $P$ is the prior distribution. To deal with this problem, we could choose a family of $P$ such that its means cover the space of $\theta$ satisfying inequality equation 7. We set

$$c = \sigma^2\big(1 + \exp\{4N/k\}\big)$$
$$P_j = \mathcal{N}\big(0, c\exp\frac{1-j}{k}\mathbb{I}_k\big)$$
$$\mathfrak{P} := \{P_j : j = 1, 2, \dots\}$$

Then the following inequality holds for a particular distribution $P_j$ with probability $1 - \delta_j$ with $\delta_j = \frac{6\delta}{\pi^2 j^2}$

$$\mathbb{E}_{\theta' \sim \mathcal{N}(\theta, \sigma^2)}\mathcal{L}_{\mathcal{D}}\big(f_{\theta'}\big) \leq \mathbb{E}_{\theta' \sim \mathcal{N}(\theta, \sigma^2)}\mathcal{L}_{\mathcal{S}}\big(f_{\theta'}\big) + \frac{1}{\beta}\left[\mathsf{KL}(Q\|P_j) + \log\frac{1}{\delta_j} + \Psi(\beta, N)\right].$$

Use the well-known equation: $\sum_{j=1}^{\infty} \frac{1}{j^2} = \frac{\pi^2}{6}$, then with probability $1 - \delta$, the above inequality holds with every $j$. We pick

$$j^* := \left\lfloor 1 - k \log \frac{\sigma^2 + \|\theta\|^2/k}{c} \right\rfloor = \left\lfloor 1 - k \log \frac{\sigma^2 + \|\theta\|^2/k}{\sigma^2(1 + \exp\{4N/k\})} \right\rfloor .$$

Therefore,

$$1 - j^* = \left\lceil k \log \frac{\sigma^2 + \|\theta\|^2/k}{c} \right\rceil$$

$$\Rightarrow \quad \log \frac{\sigma^2 + \|\theta\|^2/k}{c} \le \frac{1 - j^*}{k} \le \log \frac{\sigma^2 + \|\theta_0\|^2/k}{c} + \frac{1}{k}$$

$$\Rightarrow \quad \sigma^2 + \|\theta\|^2/k \le c \exp\left\{\frac{1 - j^*}{k}\right\} \le \exp(1/k)\left[\sigma^2 + \|\theta\|^2/k\right]$$

$$\Rightarrow \quad \sigma^2 + \|\theta\|^2/k \le \sigma_{P_{j^*}}^2 \le \exp(1/k)\left[\sigma^2 + \|\theta\|^2/k\right] .$$

Thus the KL term could be bounded as follow

$$\begin{aligned}
\mathsf{KL}(Q\|P_{j^*}) &= \frac{1}{2}\left[\frac{k\sigma^2 + \|\theta\|^2}{\sigma_{P_{j^*}}^2} - k + k \log \frac{\sigma_{P_{j^*}}^2}{\sigma^2}\right] \\
&\le \frac{1}{2}\left[\frac{k(\sigma^2 + \|\theta\|^2/k)}{\sigma^2 + \|\theta\|^2/k} - k + k \log \frac{\exp(1/k)(\sigma^2 + \|\theta\|^2/k)}{\sigma^2}\right] \\
&= \frac{1}{2}\left[k \log \frac{\exp(1/k)(\sigma^2 + \|\theta\|^2/k)}{\sigma^2}\right] \\
&= \frac{1}{2}\left[1 + k \log\left(1 + \frac{\|\theta_0\|^2}{k\sigma^2}\right)\right]
\end{aligned}$$

For the term $\log \frac{1}{\delta_{j^*}}$, with recall that $c = \sigma^2(1 + \exp(4N/k))$ and $j^* = \left\lfloor 1 - k \log \frac{\sigma^2 + \|\theta\|^2/k}{\sigma^2(1 + \exp\{4N/k\})} \right\rfloor$, we have

$$\begin{aligned}
\log \frac{1}{\delta_{j^*}} &= \log \frac{(j^*)^2 \pi^2}{6\delta} = \log \frac{1}{\delta} + \log\left(\frac{\pi^2}{6}\right) + 2\log(j^*) \\
&\le \log \frac{1}{\delta} + \log \frac{\pi^2}{6} + 2\log\left(1 + k \log \frac{\sigma^2(1 + \exp(4N/k))}{\sigma^2 + \|\theta\|^2/k}\right) \\
&\le \log \frac{1}{\delta} + \log \frac{\pi^2}{6} + 2\log\left(1 + k \log\left(1 + \exp(4N/k)\right)\right) \\
&\le \log \frac{1}{\delta} + \log \frac{\pi^2}{6} + 2\log\left(1 + k\left(1 + \frac{4N}{k}\right)\right) \\
&\le \log \frac{1}{\delta} + \log \frac{\pi^2}{6} + \log(1 + k + 4N) .
\end{aligned}$$

Hence, the inequality

$$\begin{aligned}
\mathcal{L}_{\mathcal{D}}\left(\theta', \mathcal{N}(\theta, \sigma^2 \mathbb{I}_k)\right) &\le \mathcal{L}_{\mathcal{S}}\left(\theta', \mathcal{N}(\theta, \sigma^2 \mathbb{I}_k)\right) + \sqrt{\frac{\mathsf{KL}(Q\|P_{j^*}) + \log \frac{1}{\delta_{j^*}}}{2N}} \times L \\
&\le \mathcal{L}_{\mathcal{S}}\left(\theta', \mathcal{N}(\theta, \sigma^2 \mathbb{I}_k)\right) + \frac{L}{2\sqrt{N}}\sqrt{1 + k \log\left(1 + \frac{\|\theta\|^2}{k\sigma^2}\right) + 2\log \frac{\pi^2}{6\delta} + 4\log(N + k)} \\
&\le \mathcal{L}_{\mathcal{S}}\left(\theta', \mathcal{N}(\theta, \sigma^2 \mathbb{I}_k)\right) + \frac{L}{2\sqrt{N}}\sqrt{k \log\left(1 + \frac{\|\theta\|^2}{k\sigma^2}\right) + O(1) + 2\log \frac{1}{\delta} + 4\log(N + k)} .
\end{aligned}$$

Since $\|\theta' - \theta\|^2$ is $k$ chi-square distribution, for any positive $t$, we have

$$\mathbb{P}\left(\|\theta' - \theta\|^2 - k\sigma^2 \ge 2\sigma^2\sqrt{kt} + 2t\sigma^2\right) \le \exp(-t) .$$

By choosing $t = \frac{1}{2}\log(N)$, with probability $1 - N^{-1/2}$, we have

$$\|\theta' - \theta\|^2 \leq \sigma^2 \log(N) + k\sigma^2 + \sigma^2 \sqrt{2k\log(N)} \leq k\sigma^2 \left(1 + \sqrt{\frac{\log(N)}{k}}\right)^2.$$

By setting $\sigma = \rho \times \left(\sqrt{k} + \sqrt{\log(N)}\right)^{-1}$, we have $\|\theta' - \theta\|^2 \leq \rho^2$. Hence, we get

$$\mathcal{L}_{\mathcal{S}}\left(\theta', \mathcal{N}(\theta, \sigma^2 \mathbb{I}_k)\right) = \mathbb{E}_{\theta \sim \mathcal{N}(\theta, \sigma^2 \mathbb{I}_k)} \mathbb{E}_{\mathcal{S}}\left[f_{\theta'}\right] = \int_{\|\theta' - \theta\| \leq \rho} \mathbb{E}_{\mathcal{S}}\left[f_{\theta'}\right] d\mathcal{N}(\theta, \sigma^2 \mathbb{I}) + \int_{\|\theta' - \theta\| > \rho} \mathbb{E}_{\mathcal{S}}\left[f_{\theta'}\right] d\mathcal{N}(\theta, \sigma^2 \mathbb{I})$$

$$\leq \left(1 - \frac{1}{\sqrt{N}}\right) \max_{\|\theta' - \theta\| \leq \rho} \mathcal{L}_{\mathcal{S}}(\theta') + \frac{1}{\sqrt{N}} L$$

$$\leq \max_{\|\theta' - \theta\|_2 \leq \rho} \mathcal{L}_{\mathcal{S}}(\theta') + \frac{2L}{\sqrt{N}}.$$

It follows that

$$\mathcal{L}_{\mathcal{D}}(\theta) \leq \max_{\|\theta' - \theta\| \leq \rho} \mathcal{L}_{\mathcal{S}}(\theta') + \frac{4L}{\sqrt{N}} \left[\sqrt{k\log\left(1 + \frac{\|\theta\|^2}{\rho^2}\left(1 + \sqrt{\log(N)/k}\right)^2\right)} + 2\sqrt{\log\left(\frac{N+k}{\delta}\right)} + O(1)\right].$$

Replace $\theta = \theta_i$, with probability $1 - \delta/(m+1)$ we have

$$\mathcal{L}_{\mathcal{D}}(\theta_i) \leq \mathcal{L}_{\mathcal{D}}\left(\theta'_i, \mathcal{N}(\theta_i, \sigma^2 \mathbb{I})\right) \leq \max_{\|\theta'_i - \theta_i\| < \rho} \mathcal{L}_{\mathcal{S}}(\theta'_i) + \frac{4L}{\sqrt{N}} \left[\sqrt{k\log\left(1 + \frac{\|\theta_i\|^2}{\rho^2}\left(1 + \sqrt{\log(N)/k}\right)^2\right)} + \right.$$

$$\left. 2\sqrt{\log\frac{(m+1)(N+k)}{\delta}} + O(1)\right].$$

For the loss on the ensemble classifier, we use the assumption

$$\mathbb{E}_{(x,y) \in \mathcal{D}}\left[\ell(f_\theta^{\mathrm{ens}}(x), y)\right] \leq \mathbb{E}_{\theta \sim P} \mathbb{E}_{(x,y) \in \mathcal{D}}\left[\ell(f_\theta^{\mathrm{ens}}(x), y)\right].$$

Repeating the same step of proof for $\theta$, with probability at least $1 - \delta/(m+1)$, we obtain

$$\mathcal{L}_{\mathcal{D}}(\theta) \leq \mathcal{L}_{\mathcal{D}}\left(\theta', \mathcal{N}(\theta, \sigma^2 \mathbb{I}_{mk})\right)$$

$$\leq \max_{\|\theta' - \theta\| \leq \sqrt{m}\rho} \mathcal{L}_{\mathcal{S}}(\theta') + \frac{4L}{\sqrt{N}} \left[\sqrt{mk\log\left(1 + \frac{\sum_{i=1}^m \|\theta_i\|^2}{m\rho^2}\left(1 + \sqrt{\log(N)/mk}\right)^2\right)} + \right.$$

$$\left. 2\sqrt{\log\frac{(m+1)(N+mk)}{\delta}} + O(1)\right].$$

By the convexity property of $\ell$, we have

$$\mathcal{L}_{\mathcal{D}}(\theta) \leq \mathbb{E}_{(x,y) \sim \mathcal{D}}\left[\ell\left(\frac{1}{m}\sum_{i=1}^m f_{\theta_i}(x)^{(i)}, y\right)\right] \leq \mathbb{E}_{(x,y) \sim \mathcal{D}}\left[\frac{1}{m}\sum_{i=1}^m \ell\left(f_{\theta_i}^{(i)}(x), y\right)\right]$$

$$\leq \frac{1}{m}\sum_{i=1}^m \mathcal{L}_{\mathcal{D}}(\theta_i).$$

Finally, we obtain

$$\mathcal{L}_{\mathcal{D}}(\theta) \leq \gamma \max_{\theta': \|\theta' - \theta\| \leq \sqrt{m}\rho} \mathcal{L}_{\mathcal{S}}(\theta') + \frac{1-\gamma}{m}\left[\sum_{i=1}^m \max_{\theta'_i: \|\theta'_i - \theta_i\| \leq \rho} \mathcal{L}_{\mathcal{S}}(\theta'_i)\right] + \frac{CL}{\sqrt{N}} \times$$

$$\left[m\sqrt{\log\frac{m(N+k)}{\delta}} + \sum_{i=1}^m \sqrt{k\log\left(1 + \frac{\|\theta_i\|^2}{\rho^2}(1 + \sqrt{\log(N)}/k)^2\right)} + \right.$$

$$\left. \sqrt{km\log\left(1 + \frac{\sum_{i=1}^m \|\theta_i\|^2}{m\rho^2}\left(1 + \sqrt{\log(N)/(mk)}\right)^2\right)} + O(1)\right]$$

where $C$ is an universal constant. $\qquad\square$

# B  TRAINING ALGORITHM

We present the pseudo code for our proposed method DASH as in Algorithm 1 and also provide our implementation in the anonymous link https://anonymous.4open.science/r/DASH. It is a worth-noting that we utilize the cross entropy loss with label smoothing with $\alpha = 0.1$ as the loss function $l$. Compared to the standard SAM (Foret et al., 2021), our method has one modification in the first optimization phase when we consider the diverse-aware loss $\mathcal{L}_B^{div}(\theta_i, \theta_{\neq i})$ in addition to the predictive loss $\mathcal{L}_B(\theta_i)$ in order to find the perturbed weight $\theta_i^a$. However, this process requires to calculate the gradients $\nabla_{\theta_i}\mathcal{L}_B(\theta_i)$ and $\nabla_{\theta_i}\mathcal{L}_B^{div}(\theta_i, \theta_{\neq i})$ of the two losses with respect to the same $\theta_i$ separately which consumes one more back-propagation step compared to SAM. Therefore, in practice, we alternatively consider to maximize the combined loss $\mathcal{L}_B^c(\theta_i) = \mathcal{L}_B(\theta_i) + \gamma_c\ \mathcal{L}_B^{div}(\theta_i, \theta_{\neq i})$ to find the perturbed weight $\theta_i^a$. The trade-off parameter $\gamma_c$ now replaces the perturbed radius $\rho_2$ and can be found adaptively by adjusting the strength of two gradients in the first iteration of each epoch. By using this approach, we can use the same number of back-propagation step as SAM in the first optimization phase.

---

**Input:** Training set $\mathcal{S} \triangleq \{(x_n, y_n)\}_{n=1}^N$; Loss function $l : \mathbb{R}^M \times \mathcal{Y}$; Batch size $b$; Learning rate $\eta$; Trade-off parameter $\gamma$; Perturbed radiuses $\rho_1, \rho_2$; Ensemble size $m$.
**Output:** Ensemble trained with DASH $\theta^t$
Initialize weights for $m$ base learners $\theta^0 := [\theta_i]_{i=1}^m, t = 0$;
**while** *not converged* **do**
    Sample batch $B = \{(x_1, y_1), ...(x_b, y_b)\}$;
    **for** $i \leftarrow 1$ *to* $m$ **do**
        Compute gradient $\nabla_{\theta_i}\mathcal{L}_B(\theta_i)$ of the batch's training loss;
        Compute $i$-th divergence loss $\mathcal{L}_B^{div}(\theta_i, \theta_{\neq i})$ per Eq. (4);
        Compute the perturbed weight: $\theta_i^a = \theta_i + \rho_1 \frac{\nabla_{\theta_i}\mathcal{L}_B(\theta_i)}{\|\nabla_{\theta_i}\mathcal{L}_B(\theta_i)\|} + \rho_2 \frac{\nabla_{\theta_i}\mathcal{L}_B^{div}(\theta_i, \theta_{\neq i})}{\|\nabla_{\theta_i}\mathcal{L}_B^{div}(\theta_i, \theta_{\neq i})\|}$ per Eq.
        (3);
        Update weights: $\theta_i = \theta_i - \eta\nabla_{\theta_i}\mathcal{L}_B(\theta_i^a)$;
    **end**
    $\theta^{t+1} \leftarrow [\theta_i]_i^m$;
    $t = t + 1$
**end**

**Algorithm 1:** DASH Algorithm

---

# C  EXPERIMENTAL SETTING

## C.1  BASELINES

The goal of our experiments is to evaluate the predictive performance of DASH Ensemble. We compare DASH against top ensemble learning methods consistently reported to yield high accuracies: Deep ensembles (Lakshminarayanan et al., 2017), Snapshot ensembles (Huang et al., 2017), Fast Geometric Ensemble (FGE) (Garipov et al., 2018), sparse ensembles EDST and DST (Liu et al., 2022). We use SGD optimizer and the same weight decay rate at $0.005$ for all methods including DASH. For the remaining hyper-parameters, we reuse the best settings reported in the baseline papers for the appropriate datasets. For example, Snapshot and FGE have customized learning rate schedulers as part of their proposed frameworks. The final prediction of the ensemble is obtained by taking the unweighted average of individual predictions from base learners. This strategy is applied consistently for the baselines to ensure fair comparison. While many existing works employ stronger learners such as ResNet50 or WideResNet in their experiments, it is also worth paying attention to the base architectures used in our experiments that mainly involve weak learners. This is because DASH inherits the principles of classic ensembles i.e., bagging or boosting, which aims to combine weak learners to improve the overall predictive capacity.

## C.2  DATASET SETTING

Through our experimenents, we make use of three datasets including the CIFAR10, CIFAR100 (Krizhevsky et al., 2009) and Tiny-ImageNet datasets. The CIFAR10 and CIFAR100 datasets have 50k training images and 10k testing images, with the image resolution of $32 \times 32 \times 3$. The Tiny-ImageNet datasets consists of 100k training images, 10k valiation and 10k testing images, all with

resolution of $64 \times 64 \times 3$. The CIFAR10 dataset has just 10 classes while the CIFAR100 and Tiny-ImageNet are more complex datasets with 100 classes and 200 classes, respectively. It is worth noting that, in the Tiny-ImageNet dataset, we evaluate on the validation set instead of testing set, which is a common practice in the literature.

We follow the standard data pre-processing schemes for the CIFAR10 and CIFAR100 datasets that consists of zero-padding with 4 pixels on each side, random crop, horizon flip and normalization. For the Tiny-Imagenet dataset, we apply resize and random crop operations to change the resolution to $224x224x3$ as similar as the Imagenet dataset.

### C.3  TRAINING SETTING

The training configuration shared between our method and the baselines involves model training for 200 epochs using SGD optimizer with weight decay of 5e-3 and momentum 0.9. We use the learning rate scheduler, starting with learning rate 0.1 and changing at epoch 60th, 120th and 160th with scale 0.2 as suggested in the project https://github.com/davda54/sam.

## D  ADDITIONAL EXPERIMENTS

### D.1  EVALUATION OF UNCERTAINTY ESTIMATION

We would like to provide the complete experimental results with all **six** Uncertainty Estimation metrics on the CIFAR10 dataset (Tables 6, 7, 8), the CIFAR100 dataset (Tables 9, 10, 11) and the Tiny-ImageNet dataset (Table 5).

On evaluation of the predictive performance, as reported in Section 4.3 in the main paper, our proposed method DASH consistently and significantly outperforms all baselines across all datasets and architectures. Unlike Fast Geometric, Snapshot, or EDST methods, which are limited to homogeneous ensemble setting, our DASH is a general method capable on either homogeneous or heterogeneous ensemble.

On evaluation of the uncertainty estimation capability, in addition to the result on the Tiny-Imagenet dataset that has been reported in Section 4.4 in the main paper, Tables 6, 7, 8 show the results on the CIFAR10 dataset, with R10x5, R18x3 and RME architectures, respectively, while Tables 9, 10, 11 show the results of the same architectures on the CIFAR100 dataset. As similar as the observation on the Tiny-Imagenet dataset, it can be seen from the results on the CIFAR10 and CIFAR100 datasets that our proposed method DASH achieves the best performance on the five UE metrics, except for the ECE metric. In comparison to the Deep Ensemble, our method achieves much better performance on the heterogeneous setting (i.e., RME architecture) as seen from Table 8 or Table 11. Overall, the experimental results demonstrate the superiority of our proposed DASH method over other baseline methods for both predictive performance and UE capabilities across all datasets and architectures.

Table 5: Evaluation on the Tiny-ImageNet dataset with R18x3 architecture.

| | Accuracy ↑ | NLL ↓ | Brier ↓ | ECE ↓ | Cal-Brier ↓ | Cal-AAC ↓ | Cal-NLL ↓ |
|---|---|---|---|---|---|---|---|
| Deep Ensemble | 65.9 | 1.400 | 0.452 | **0.110** | 0.453 | 0.210 | 1.413 |
| Fast Geometric | 61.8 | 1.548 | 0.501 | 0.116 | 0.499 | 0.239 | 1.544 |
| Snapshot | 62.2 | 1.643 | 0.505 | 0.118 | 0.501 | 0.237 | 1.599 |
| EDST | 62.3 | 1.581 | 0.496 | 0.115 | 0.495 | 0.235 | 1.548 |
| DST | 61.9 | 1.525 | 0.499 | **0.110** | 0.500 | 0.239 | 1.536 |
| SGD | 62.3 | 1.999 | 0.601 | 0.283 | 0.518 | 0.272 | 1.737 |
| SAM | 66.1 | 1.791 | 0.563 | 0.297 | 0.469 | 0.242 | 1.484 |
| DASH (Ours) | **69.9** | **1.379** | **0.447** | 0.184 | **0.407** | **0.204** | **1.213** |

Table 6: Evaluation on the CIFAR10 dataset with R10x5 architecture.

| | Accuracy ↑ | NLL ↓ | Brier ↓ | ECE ↓ | Cal-Brier ↓ | Cal-AAC ↓ | Cal-NLL ↓ |
|---|---|---|---|---|---|---|---|
| Deep Ensemble | 92.7 | 0.226 | 0.107 | **0.053** | 0.091 | 0.108 | 0.272 |
| Fast Geometric | 92.5 | 0.555 | 0.261 | 0.113 | 0.251 | 0.144 | 0.531 |
| Snapshot | 93.6 | 0.202 | 0.095 | 0.048 | 0.083 | **0.107** | 0.249 |
| EDST | 92.0 | 0.245 | 0.118 | 0.057 | 0.122 | 0.112 | 0.301 |
| DST | 93.2 | 0.211 | 0.099 | 0.049 | 0.102 | 0.108 | 0.261 |
| SGD | 95.1 | 0.277 | 0.096 | 0.143 | 0.078 | 0.108 | 0.264 |
| SAM | 95.4 | 0.257 | 0.087 | 0.136 | 0.073 | **0.107** | 0.268 |
| DASH (Ours) | **95.7** | **0.244** | **0.084** | 0.134 | **0.067** | **0.107** | **0.248** |

Table 7: Evaluation on the CIFAR10 dataset with R18x3 architecture.

| | Accuracy ↑ | NLL ↓ | Brier ↓ | ECE ↓ | Cal-Brier ↓ | Cal-AAC ↓ | Cal-NLL ↓ |
|---|---|---|---|---|---|---|---|
| Deep Ensemble | 93.7 | 0.197 | 0.091 | 0.047 | 0.079 | **0.107** | 0.273 |
| Fast Geometric | 93.3 | 0.257 | 0.108 | 0.055 | 0.087 | 0.108 | 0.261 |
| Snapshot | 94.8 | 0.201 | 0.082 | 0.043 | 0.071 | 0.108 | 0.270 |
| EDST | 92.8 | 0.231 | 0.110 | 0.054 | 0.113 | 0.110 | 0.281 |
| DST | 94.7 | 0.172 | 0.080 | **0.042** | 0.083 | **0.107** | 0.253 |
| SGD | 95.2 | 0.249 | 0.083 | 0.120 | 0.076 | 0.108 | 0.282 |
| SAM | 95.8 | 0.229 | 0.074 | 0.120 | 0.067 | **0.107** | 0.261 |
| DASH (Ours) | **96.7** | **0.215** | **0.065** | 0.124 | **0.056** | **0.107** | **0.250** |

Table 8: Evaluation on the CIFAR10 dataset with RME architecture.

| | Accuracy ↑ | NLL ↓ | Brier ↓ | ECE ↓ | Cal-Brier ↓ | Cal-AAC ↓ | Cal-NLL ↓ |
|---|---|---|---|---|---|---|---|
| Deep Ensemble | 89.0 | 0.905 | 0.391 | 0.431 | 0.153 | 0.126 | 0.395 |
| DST | 93.4 | 0.209 | 0.101 | **0.058** | 0.102 | 0.109 | 0.282 |
| SGD | 92.6 | 0.328 | 0.128 | 0.136 | 0.113 | 0.112 | 0.317 |
| SAM | 93.8 | 0.310 | 0.112 | 0.145 | 0.094 | 0.110 | 0.280 |
| DASH (Ours) | **95.2** | **0.276** | **0.095** | 0.151 | **0.075** | **0.106** | **0.236** |

Table 9: Evaluation on the CIFAR100 dataset with R10x5 architecture.

| | Accuracy ↑ | NLL ↓ | Brier ↓ | ECE ↓ | Cal-Brier ↓ | Cal-AAC ↓ | Cal-NLL ↓ |
|---|---|---|---|---|---|---|---|
| Deep Ensemble | 73.7 | 0.973 | 0.365 | **0.101** | 0.329 | 0.162 | 0.870 |
| Fast Geometric | 63.2 | 1.926 | 0.658 | 0.213 | 0.606 | 0.324 | 1.723 |
| Snapshot | 72.8 | 1.072 | 0.382 | 0.112 | 0.338 | 0.165 | 0.929 |
| EDST | 68.4 | 1.142 | 0.427 | 0.112 | 0.427 | 0.207 | 1.151 |
| DST | 70.8 | 1.064 | 0.393 | 0.103 | 0.396 | 0.189 | 1.076 |
| SGD | 75.9 | 1.502 | 0.522 | 0.400 | 0.346 | 0.174 | 1.001 |
| SAM | 77.7 | 1.302 | 0.460 | 0.357 | 0.321 | 0.164 | 0.892 |
| DASH (Ours) | **80.8** | **0.864** | **0.316** | 0.180 | **0.271** | **0.144** | **0.684** |

## D.2 EVALUATION ON ADVERSARIAL ROBUSTNESS

In this section, our goal is to evaluate the adversarial robustness of our proposed method against adversarial attacks. To achieve this, we conducted experiments on the CIFAR10 dataset using the R18x3 architecture and employed the PGD attack (Madry et al., 2017), which is considered the standard adversarial attack for evaluating robustness. Specifically, we set the number of attack steps to $k = 10$, step size to $\eta = 1/255$, and varied the change in perturbation size $\epsilon$ from $1/255$ to $6/255$.

Table 10: Evaluation on the CIFAR100 dataset with R18x3 architecture.

| | Accuracy ↑ | NLL ↓ | Brier ↓ | ECE ↓ | Cal-Brier ↓ | Cal-AAC ↓ | Cal-NLL↓ |
|---|---|---|---|---|---|---|---|
| Deep Ensemble | 75.4 | 0.927 | 0.342 | **0.095** | 0.308 | 0.155 | 0.822 |
| Fast Geometric | 72.3 | 1.12 | 0.394 | 0.124 | 0.344 | 0.169 | 0.950 |
| Snapshot | 75.7 | 1.011 | 0.347 | 0.111 | 0.311 | 0.153 | 0.903 |
| EDST | 69.6 | 1.125 | 0.412 | 0.106 | 0.412 | 0.197 | 1.123 |
| DST | 70.4 | 1.228 | 0.419 | 0.140 | 0.405 | 0.194 | 1.153 |
| SGD | 78.9 | 1.225 | 0.389 | 0.285 | 0.304 | 0.156 | 0.919 |
| SAM | 80.1 | 1.080 | 0.356 | 0.261 | 0.285 | 0.151 | 0.808 |
| DASH (Ours) | **82.2** | **0.892** | **0.300** | 0.196 | **0.255** | **0.138** | **0.679** |

Table 11: Evaluation on the CIFAR100 dataset with RME architecture.

| | Accuracy ↑ | NLL↓ | Brier ↓ | ECE ↓ | Cal-Brier ↓ | Cal-AAC ↓ | Cal-NLL ↓ |
|---|---|---|---|---|---|---|---|
| Deep Ensemble | 62.7 | 2.137 | 0.699 | 0.401 | 0.433 | 0.209 | 1.267 |
| DST | 71.7 | 1.056 | 0.393 | **0.111** | 0.393 | 0.187 | 1.066 |
| SGD | 72.6 | 1.559 | 0.531 | 0.350 | 0.403 | 0.201 | 1.192 |
| SAM | 76.4 | 1.439 | 0.501 | 0.377 | 0.347 | 0.177 | 1.005 |
| DASH (Ours) | **78.7** | **0.969** | **0.342** | 0.202 | **0.298** | **0.151** | **0.764** |

While it is widely recognized in the Adversarial Machine Learning literature that strong attacks are required to truly challenge defense methods (i.e., PGD attack with more than 200 attack steps with a perturbation size of $\epsilon = 8/255$), we chose a weaker attack for our experiments. This decision was based on the fact that all methods we evaluated were not specifically designed to enhance adversarial robustness, and therefore may not perform well against a stronger attack.

It can be seen from Figure 4a that our DASH achieves better adversarial robustness than all baselines on the R18x3 architecture. More specifically, our method consistently outperforms SGD by around 3% across different levels of $\epsilon$. While there is a huge drop of adversarial robustness on SAM when the attack becomes stronger (i.e., 61.28% with $\epsilon = 1/255$ and 27.61% with $\epsilon = 2/255$), our method is more robust with a smaller drop (i.e., 65.53% with $\epsilon = 1/255$ and 42.23% with $\epsilon = 2/255$). On the R10x5 architecture, our method still outperforms SGD and SAM across all levels of attack strength. However, it can be observed that our DASH achieves a lower performance than DST and EDST methods if the perturbation size $\epsilon \geq 2/255$ as shown in Figure 4b. While our method does not specifically target improving adversarial robustness, the superior performance we achieve on the R18x3 architecture suggests that our principle of considering sharpness-aware and diverse-aware mechanisms could be a promising direction for addressing this issue.

### D.3 HYPER-PARAMETER SENSITIVITY

In this section, we investigate the effect of the hyper-parameter $\gamma$ on the performance of our method by tuning it over the range of [0, 1]. Recall that $\gamma = 0$ means that we seek flatness on all individual base classifiers but not the entire ensemble model, while $\gamma = 1$ means that we seek flatness on the entire aggregated ensemble model only. We conduct the experiment on the CIFAR100 dataset with R10x5 architecture and report results on Table 12. It can be seen that our method achieves the best performance in both generalization and uncertainty estimation aspects when $\gamma = 0$ and there is a significant drop of 1.8% in accuracy when $\gamma = 1$. This result suggests that combining individual flattened base classifiers can lead to better generalization performance than seeking flatness on the entire ensemble model. In our experiments, we set $\gamma = 0$ as the default setting.

### D.4 ANALYSIS OF THE ENSEMBLE SIZE

In this section, our aim is to examine the impact of ensemble size, that is, the number of base classifiers, on the final performance. We performed an experiment on the CIFAR100 dataset by varying the ensemble size from two to seven base classifiers, in which each base classifier is a ResNet10 model. The results of this experiment are presented in Table 13 and Figure 5. Ensemble learning theory suggests that the generalization capacity of an ensemble improves with the number of base classifiers, assuming the base classifiers exhibit diversity. Figure 5 demonstrates that the

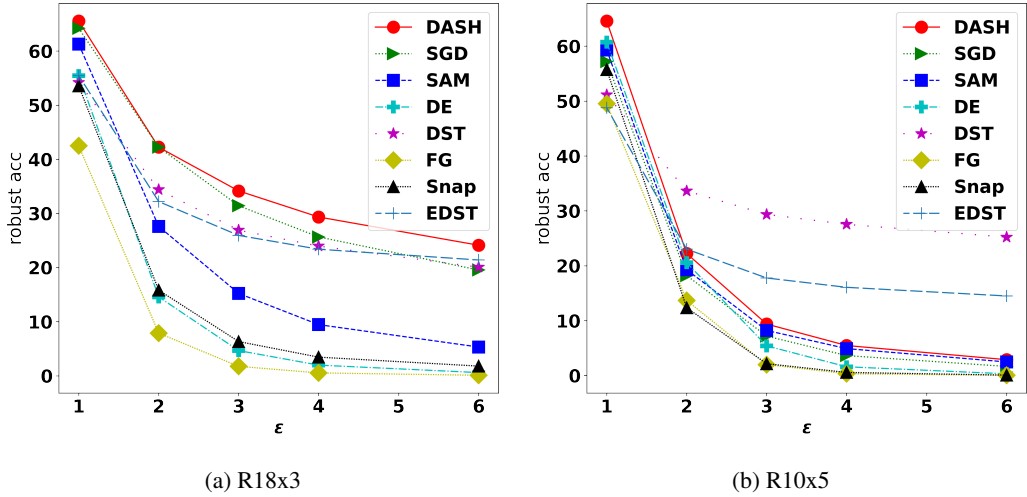

(a) R18x3            (b) R10x5

Figure 4: Evaluation on Adversarial Robustness. The x-axis denotes the perturbation size $\epsilon$ (*255).

Table 12: Evaluation of ensemble accuracy under various the trade-off parameters $\gamma$. ↑ Higher is better. ↓ Lower is better.

|  | Accuracy ↑ | NLL ↓ | Brier ↓ | ECE ↓ |
|---|---|---|---|---|
| $\gamma = 0.0$ | **80.84** | **0.86** | **0.32** | **0.18** |
| $\gamma = 0.2$ | 80.48 | 0.97 | 0.35 | 0.23 |
| $\gamma = 0.5$ | 80.42 | 0.95 | 0.34 | 0.22 |
| $\gamma = 0.8$ | 79.81 | 1.08 | 0.38 | 0.29 |
| $\gamma = 1.0$ | 78.86 | 1.12 | 0.40 | 0.28 |

ensemble accuracy increases linearly with the number of base classifiers, with a 1.7% improvement in ensemble accuracy when increasing the number of base classifiers from 2 to 6. Furthermore, our method's uncertainty estimation capability benefits from a larger ensemble size, as evidenced by the improvements in all three UE metrics shown in Table 13. Interestingly, we also observed that the performance of the base classifiers, as measured by the average accuracy metric in Table 13, also improves when working in collaboration with a larger number of base classifiers, with an accuracy improvement of 0.4%. However, we noted that the benefits of a larger ensemble appear to reach a saturation point when the number of base classifiers exceeds 6.

### D.5 Contribution of each component

In this section, our objective is to assess the impact of each component by comparing the performance of two variants: DASH and DASH$^F$, which is our method with flat seeking mode only. To conduct the experiment, we used the CIFAR10 and CIFAR100 datasets with RME architecture, and

Table 13: Evaluation of ensemble performance on CIFAR100 with different number of base classifiers. *Avg.* denotes the average accuracy value all base learners.

|  | Accuracy ↑ | Avg. Accuracy ↑ | Cal-Brier ↓ | Cal-AAC ↓ | Cal-NLL ↓ |
|---|---|---|---|---|---|
| R10x2 | 79.19 | 77.60 | 0.289 | 0.792 | 0.751 |
| R10x3 | 79.86 | 77.49 | 0.280 | 0.147 | 0.715 |
| R10x4 | 80.71 | 77.77 | 0.272 | 0.143 | 0.691 |
| R10x5 | 80.84 | 77.94 | 0.267 | 0.142 | 0.676 |
| R10x6 | 80.89 | 77.98 | 0.268 | 0.142 | 0.677 |
| R10x7 | 80.89 | 77.88 | 0.268 | 0.142 | 0.673 |

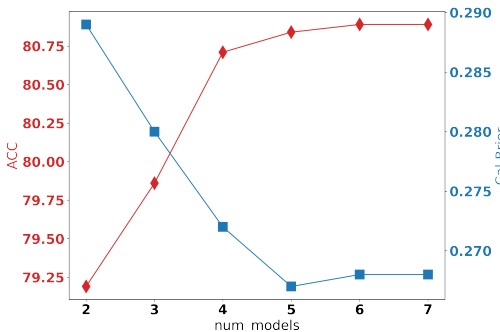

Figure 5: Evaluation of different of number of ResNet10 base classifiers on the CIFAR100 dataset.

the results are presented in Table 14. We observed that DASH$^F$ outperforms the standard SGD method by a significant margin when using the flat seeking mode only. The performance improvement is remarkable, with a gap of 1.72% and 3.73% on the CIFAR10 and CIFAR100 datasets, respectively. This enhancement can be attributed to the improvement of each single base classifier. The ensemble can achieve better generalization performance by combining these classifiers. In particular, the average accuracy of all base classifiers with DASH$^F$ is 93.21%, which is 5.07% higher than that achieved with the SGD method. However, in terms of ensemble diversity, measured by the Log-Determinant metric, DASH$^F$'s base classifiers are less diverse than those of SGD. Specifically, on the same CIFAR100 dataset, SGD obtains a LD score of -16.88, while that of DASH$^F$ is only -19.47, which is a 15.3% relatively lower. The lower LD score indicates that the predictions of the base classifiers on DASH$^F$ have a higher similarity than those on SGD. Consequently, in some hard negative samples, the predictions of all base classifiers fall into similar incorrect patterns, and the final ensemble prediction becomes incorrect. On the other hand, when comparing between DASH and DASH$^F$, it can be observed that, DASH obtains a higher LD score in both datasets, while also improves the average performance of the base classifiers. As consequent, DASH improves over DASH$^F$ by 0.84% 2.44% on the CIFAR10 and CIFAR100, respectively.

Table 14: Ablation study on the contribution of each component on the CIFAR10 (C10) and CIFAR100 (C100) datasets with RME architecture. DASH$^F$ represents our method with flat seeking mode only.

|  |  | Accuracy ↑ | LD ↑ | D ↑ | Avg. Accuracy ↑ |
|---|---|---|---|---|---|
| | SGD | 92.61 | -24.7 | **0.149** | 88.14 |
| C10 | DASH$^F$ | 94.33 | -25.8 | 0.034 | 93.21 |
| | DASH | **95.17** | **-23.3** | 0.068 | **93.41** |
| | SGD | 72.55 | **-16.88** | **0.853** | 38.09 |
| C100 | DASH$^F$ | 76.28 | -19.47 | 0.123 | 73.38 |
| | DASH | **78.72** | -18.92 | 0.237 | **74.69** |

### D.6 ANALYSIS OF THE CHOICE OF THE HYPER-PARAMETER

As mentioned in Section 3.3, in default we choose $\rho_1 = \rho_2$ for simplicity, with the value of 2.0 as recommended in Kwon et al. (2021). To understand more the effect of hyper-parameters on the performance of our method, in this experiment, we choose different $\rho_2$ values while fixing $\rho_1$. It can be seen from Table 15 that though fine-tuning helps slightly improve our current performance (at $\rho_1 = \rho_2$), simple hyper-parameter setting that works well is an advantage of our approach.

### D.7 EXPERIMENTS WITH WIDERESNET

Table 16 reports the predictive accuracy of all methods on an ensemble of three WideResNet28-10 models. It can be seen that our method consistently outperforms all baselines across different choices of model architectures and datasets.

Table 15: Analysis of the choice of the hyper-parameter. *Current setting

| $\rho_2/\rho_1$ | 0 | 0.1 | **1.0***  | 2.0 | 3.0 |
|---|---|---|---|---|---|
| Acc.↑ | 80.07 | 82.14 | 82.19 | **82.44** | 82.25 |
| Cal-Brier↓ | 0.285 | 0.255 | 0.255 | **0.246** | 0.248 |

Table 16 also reports the training time measured in seconds per epoch on a single GPU which shows that the time efficiency of our method is comparable to many baselines.

Table 16: Additional experiment with WideResNet28-10 architecture

| | CIFAR10 | | CIFAR100 | |
|---|---|---|---|---|
| | Acc.↑ | Time (s)↓ | Acc.↑ | Time (s)↓ |
| Deep Ensemble | 91.61 | 360 | 70.92 | 560 |
| Fast Geometric | 91.04 | 308 | 72.63 | 540 |
| Snapshot | 94.63 | 216 | 76.52 | 360 |
| EDST | 96.30 | **84** | 82.20 | **84** |
| DST | 96.30 | 252 | 83.30 | 294 |
| SGD | 96.09 | 190 | 80.23 | 195 |
| SAM | 96.61 | 390 | 81.95 | 400 |
| DASH (Ours) | **97.21** | 460 | **84.09** | 470 |

