# OpenReview forum: "Diversity-Aware Agnostic Ensemble of Sharpness Minimizers"
_ICLR.cc/2024/Conference — ICLR 2024 Conference Withdrawn Submission_

### Official Review · Reviewer_XYNF · 2023-10-13

**Soundness:** 2 fair
**Presentation:** 2 fair
**Contribution:** 2 fair
**Rating:** 3
**Confidence:** 4

**Summary:**

The work analyzes the possible synergy between sharpness-aware minimization and diversity regularization when training ensemble of networks simultaneously.
Indeed, those two strategies hgave been to help when cnsidering one single model, but this paper shows (in Appendix) that more flateness actually reduces diversity across members.
They propose to adapt SAM by integrating a divertsity regularization when commuting the upward gradient term.
The authors back their approach with some theoretical foundations and empirical evidence on medium-scale datasets.

**Strengths:**

- How to combine flatness and diversity in ensembling is an interesting research direction.
- the experiments use a good methodology, with temperature scaling for calibration, and appropriate diversity measures (though they are only used in the Appendix).
- The idea to use diversity regularization in the upward of SAM (rather than more trivially in the downward) is interesting.

**Weaknesses:**

- I think that Theorem 1 does not help understanding the paper, and I am not convinced by its significance, as explained below.

    * First, the ensembling nature of $\theta$ is actually very superficial in the proof; it's simply dealt by considering that ensembling performs in average better that its members, thanks to the convexity of the loss. And thus the long computations (from page 13 to the middle of page 16) can actually be made for a single model. Actually, is this novel or is it from an existing paper? Then authors state "For the first time, we connect the sharpness-aware minimization with ensemble learning"; but actually your proof is not specific to ensemble, as you actually simply use the convexity of the loss to upper bound the ensemble by its members. This also questions the tightness of the bound.
    * The underlying assumptions are only stated along the proof, and not before.
    * Should'nt it be Q (instead of P) in the Equation from Page 16 just after "For the loss on the ensemble classifier, we use the assumption"?
    * The last line of the proof is not explicited. The "Finally, we obtain" is not clear; could you please clarify which equations are you combining to obtain the final equation?
    * Also, The C universal constant is not defined.
    * Also, please keep the equations numbered when possible, it facilitates reviewing.
    * More specifically, authors state that "The trade-off parameter $\gamma$ signifies the levels of sharpness-aware enforcement for the ensemble model alone and its base learners themselve". This coefficient gamma should be better introduced, to clarify how it appears in the proof (I suspect this is related to the unclear "Finally, we obtain"). Therefore it is not clear how you change it in Figure 1 and Appendix D.3. I speculate that for $\gamma=1$, you actually compute the upward gradient on the ensemble rather than each member, which would reduce performance because you would give the same upward gradient for each member and thus reducing diversity.

- The explanations to explain why applying diversity in the upward are hand-wavy and not compelling. The arguments (based on Taylor expansion) between the congruence (do you mean (cosine) similarity?) between the gradients and the diversity loss are not clear.

- The training paradigm would be complex to scale, as it requires simultaneous training of multiple deep networks.

- The experiments are only on medium scale datasets, and the performances are far from state-of-the-art, despite using very costly ensembling solutions. In case of computational limitation, I would suggest fine-tuning scenarios on larger datasets rather than training from scratch.

- A missing and important baseline is one that includes diversity regualrization in the downward of the SAM rather than the upward.

- Appendix D.5. should be included as an important section of the main paper, and needs to be further analyzed. Also note that in terms of disagreement (D), your strategy actually remains very far from standard SGD in Table 14.

- "To the best of our knowledge, we are the first to explore the connection between ensemble diversity and loss sharpness". No actually, "Diverse Weight Averaging for Out-of-Distribution Generalization" previously showed that SAM flattens loss landscape but actually reduces diversity across different models.

- Minors
    * The anonymous github is expired.
    * Authors state "The reason is that we aim to diversify the base learners without interfering the their performance on predicting ground-truth labels. " This was first used in "Improving adversarial robustness via promoting ensemble diversity", please cite accordingly.
    * Tables do not fit in pages in Appendix

**Questions:**

- In Table 1, what is the difference between SGD and Deep Ensemble ?
- What is the difference between SAM in Table 1 and DASHF in Appendix D.5? Why scores are not matching between Table 1 and Tabel 14?
- Did you use the same initialization across runs ? if yes, why, as it would reduce diversity ? if no, I don't understand why they would end up being on the same loss region?

---

### Official Review · Reviewer_XaPX · 2023-10-28

**Soundness:** 2 fair
**Presentation:** 2 fair
**Contribution:** 1 poor
**Rating:** 3
**Confidence:** 4

**Summary:**

This work studies the integration of the deep ensemble and sharpness minimization methods. It proposes to add a KL-divergence-based diversity-encouraging term into the loss function to encourage diversity while ensuring flatness. It provides classification accuracy and uncertainty estimation evaluation on CIFAR10, CIFAR100, and Tiny-ImageNet.

**Strengths:**

The experimental details and the results are well-presented in the tables of the main paper and the appendix.

**Weaknesses:**

* **(Unclear motivation and underdeveloped problem statement)**
This paper aims to integrate two previously established and effective methods and its novelty appears to be notably constrained.

* **(Figure 2 and confusing claim)**
I am confused if Figure 2 reflects any real empirical or theoretical insights to support the claim that the diversity is reduced because different models use the same mini-batch $B$ in the normalized gradients.
A question arises: deep ensemble trains the models independently, in that case, how could the two independent models use the same ascent gradient computed from the same mini-batch data?

* **(Lack of theoretical proof)**
The theoretical work only shows that combining SAM and deep ensembling can provide a reduced generalization bound. The main contribution of using the KL divergence term is not supported by any proof.

* **(The deep ensemble baseline accuracy is unusually low)** The SGD deep ensemble baseline (first row of Table 1) uses 3 ResNet18 to obtain **93.7%** test accuracy on CIFAR10, and **75.4%** on CIFAR100. However, in Table 1 of [1], a single ResNet18 with SGD can obtain **95.41%** on CIFAR10 and **78.17%** on CIFAR100. Therefore, the validity of the reported baseline results needs to be clarified.


* **(Unfair baseline comparison)** The paper mentions they use SGD optimizer for baseline methods (DST, EDST, Snapshot).
However, their proposed method DASH uses SAM as the default optimizer.
The evaluation can be more fair if all of the baseline ensembles can be combined with SAM and the hyperparameter $\rho$ can be carefully tuned. Then we can know that DASH is an effective ensemble method compared to the previous ensembling method instead of observing the improvement from better optimizer SAM.

* **(Lack of baseline comparison)**
The proposed DASH method is only compared with a few efficient ensemble methods (like Snapshot, FGE, and EDST) without the need to train multiple runs.
However, the proposed DASH indeed needs to be trained multiple times. The evaluation can be more solid if compared more with methods that also need multiple training runs(e.g. [1]).

* **(Limited evaluation of small models and small datasets)**. The author claims they only studied small models because their method inherits the "principles of classic ensembles (bagging or boosting)". I don't think it is a valid claim since a lot of NN ensembling work also grounds their method on the classic ensemble principles and works well in medium and large models (e.g. WideResNet). Also, relative to larger datasets ImageNet, the WRN, and ResNet50 can be considered weak learners.

* **(Missing baseline hyperparameter settings)** For results of Table 1, 2, the hyperparameter ($\rho$) range of SAM is not provided, it is possible the SAM baseline is not carefully tuned to show the improvement of the DASH method.

* **(minor) The provided code link is expired.**


[1] [Du et al.](https://arxiv.org/abs/2110.03141) "Efficient sharpness-aware minimization for improved training of neural networks."

[2] [Wenzel et al](https://arxiv.org/abs/2006.13570) "Hyperparameter ensembles for robustness and uncertainty quantification."

**Questions:**

* In Tables 1 and 2, authors compared their method with EDST and DST, which are sparse ensembling methods. Is the comparison fair here since DST models have a lower number of parameters? Therefore, is it possible to apply DASH to sparse models with the same sparsity as the DST/EDST method?

* Is it possible to provide commonly used WideResNet or ImageNet scale experiments to support the scalability of the method?

* Literature [1] shows that explicitly adding diversity-encouraging terms to loss function will hurt the ensembling performance while this work observes the improvement, is there possibly an explanation for the difference in the observations?




[1] [Abe et al](https://arxiv.org/abs/2302.00704) "Pathologies of Predictive Diversity in Deep Ensembles."

**Details Of Ethics Concerns:**

No ethics concerns.

---

### Official Review · Reviewer_xVVh · 2023-10-29

**Soundness:** 3 good
**Presentation:** 3 good
**Contribution:** 2 fair
**Rating:** 6
**Confidence:** 2

**Summary:**

This paper proposes Diversity-aware Agnostic Ensemble of Sharpness Minimizers (DASH). This method combines two broad approaches for achieving better generalization: ensembling and seeking flat minima.

**Strengths:**

The motivation behind the proposed method is clear: there are two distinct approaches that both achieve better generalization, and we would like to have a single objective that does well on both.

The paper analyzes how optimizing the additional diversity term affects training dynamics. The experimental evaluation is solid, and DASH achieves consistent performance gains over an ensemble of SAM models, among other points of comparison.

**Weaknesses:**

It would have been good if the paper compared the trained models not only on the ensemble performance, but in terms of the individual models.
1. How do the individual models compare in terms of performance? Do the average SAM and average DASH models perform similarly, with the only difference being diversity among the models?
2. How different are within-ensemble models? You could measure pairwise disagreement or even your diversity objective. Are DASH models empirically more diverse?

(minor) the bright green links are somewhat jarring.
(minor) Figures 1-3 use space inefficiently.

**Questions:**

How does the proposed method compare to standard deep ensembles in runtime / computational cost or implementation complexity? Particularly, I think the synchronous communication between models for the divergence loss can introduce difficulties.

---

### Official Review · Reviewer_UbbB · 2023-10-30

**Soundness:** 2 fair
**Presentation:** 1 poor
**Contribution:** 2 fair
**Rating:** 3
**Confidence:** 3

**Summary:**

This study delves into two commonly employed techniques for enhancing the generalization of deep neural networks: deep ensembles and sharpness-aware minimization. The proposed DASH algorithm combines aspects of ensemble diversity and sharpness-awareness, which are fundamental to the previously mentioned methods.

**Strengths:**

1. Connecting deep ensembles and sharpness-aware minimization in the perspective of generalization is a valuable research direction.

2. The authors present the experimental results using a range of evaluation metrics and consider the evaluation framework proposed by Ashukha et al. (2020) for assessing ensemble models.

**Weaknesses:**

1. The relation between the upper bound presented in Theorem 1 and the proposed DASH algorithm appears weak. The insight provided by Theorem 1 is that the generalization error can be upper-bounded by a weighted sum of the SAM losses for individual ensemble members and the SAM loss for the entire ensemble model. However, it appears that the diversification term proposed by the DASH algorithm is unrelated to this upper bound.

2. There is a lack of experimental evidence to support the occurrence of phenomena similar to what is depicted in conceptual Figures 2 and 3 when applying sharpness-aware minimization in ensemble learning. Besides, it appears improbable that ensemble members, beginning with different initializations, would converge to the same basin (Fort et al., 2019).

3. The proposed algorithm simultaneously trains $m$ parameters, which has limitations in terms of scalability (i.e., we should load $m$ model copies and corresponding computational graphs to memory). Furthermore, while the training method where multiple model copies repel each other resembles the POVI approaches involving repulsion (e.g., D’Angelo and Fortuin, 2021), this connection remains not addressed in the paper.

4. There are no standard deviations present in the experimental results.

---
Fort et al., 2019, Deep Ensembles: A Loss Landscape Perspective.
D’Angelo and Fortuin, 2021, Repulsive Deep Ensembles are Bayesian.

**Questions:**

1. What specifically does “smooth out” mean here?
    > Meanwhile, deep ensembles (i.e., ensembles of deep neural networks) are found to be able to “smooth out” the highly non-convex loss surface, resulting in a better predictive performance.

2. Section 3.3 did not provide specific experimental results, and I only found a conceptual diagram in Figure 2. My question is whether training each member of a deep ensemble with SAM indeed results in them ending up in the same low-loss region, thus diminishing ensemble diversity. Do you have any concrete experimental results regarding this point (e.g., visualizing loss surfaces)?
    > By encouraging the base learners to move closer to flat local minima, we however observe that under this sharpness-minimization scheme, the networks converge to low-loss tunnels that are close to one another, thereby compromising ensemble diversity (See Section 3.3).

3. Why does the usage of such initialization techniques reduce the ensemble diversity? In practice, we can create high-performing ensembles of deep neural networks with random initializations using the mentioned initializers.
    > This may stem from the usage of well-known initialization techniques (e.g., He initializer (He et al., 2015) or Xavier initializer (Glorot & Bengio, 2010)), making the initial base models $\theta_{i}$, $i=1,\dots,m$ significantly less diverse.

4. I understand NLL, Brier, ECE, Cal-Brier, and Cal-NLL, but what is Cal-AAC?

5. Theorem 1 suggests that the upper bound becomes looser as the ensemble size $m$ increases. Could you provide some insight into the reasons behind this?